# Chemically Denatured Structures of Porcine Pepsin using Small-Angle X-ray Scattering

**DOI:** 10.3390/polym11122104

**Published:** 2019-12-14

**Authors:** Yecheol Rho, Jun Ha Kim, Byoungseok Min, Kyeong Sik Jin

**Affiliations:** 1Chemical Analysis Center, Korea Research Institute of Chemical Technology, 141, Gajeong-ro, Yuseong-gu, Daejeon 34114, Korea; rhoyc@krict.re.kr; 2Pohang Accelerator Laboratory, Pohang University of Science and Technology, 80 Jigokro-127-beongil, Nam-Gu, Pohang, Kyungbuk 37673, Korea; junhak@postech.ac.kr (J.H.K.); moya122@postech.ac.kr (B.M.)

**Keywords:** Porcine pepsin, urea, denaturation, Raman spectroscopy, small-angle X-ray scattering (SAXS)

## Abstract

Porcine pepsin is a gastric aspartic proteinase that reportedly plays a pivotal role in the digestive process of many vertebrates. We have investigated the three-dimensional (3D) structure and conformational transition of porcine pepsin in solution over a wide range of denaturant urea concentrations (0–10 M) using Raman spectroscopy and small-angle X-ray scattering. Furthermore, 3D GASBOR ab initio structural models, which provide an adequate conformational description of pepsin under varying denatured conditions, were successfully constructed. It was shown that pepsin molecules retain native conformation at 0–5 M urea, undergo partial denaturation at 6 M urea, and display a strongly unfolded conformation at 7–10 M urea. According to the resulting GASBOR solution models, we identified an intermediate pepsin conformation that was dominant during the early stage of denaturation. We believe that the structural evidence presented here provides useful insights into the relationship between enzymatic activity and conformation of porcine pepsin at different states of denaturation.

## 1. Introduction

In general, the biological functions of proteins require the correct folding and sufficient stability. However, protein stability is strongly influenced by the type of solvent, ionic strength, and protein concentration. Protein folding is the process by which a peptide chain assumes its functional shape or conformation. All protein molecules are heterogeneous unbranched chains of amino acids. By coiling and folding into a specific three-dimensional (3D) shape, they can perform their biological functions. The reversal of this process is known as denaturation, whereby a native protein loses its functional conformation and becomes an amorphous and non-functional amino acid chain [1]. Studying the structural dynamics of denatured proteins can provide insights into the physical and chemical principles that govern protein folding. The unfolded state is frequently viewed as unstructured and featureless and is thus described as a random-coil [2]. To fully understand the conformational behavior of a protein, it is necessary to define not only the structure of its native state but also those of various denatured states [3,4]. Recent studies have revealed the biological significance of denatured states in processes such as aggregation [5,6,7], chaperone binding [8,9], and membrane transport [10,11]. A variety of denatured states have also been identified that differ in their overall dimensions and the extent of residual secondary and tertiary structures [12,13,14]. In particular, recent studies have shown that heat- and chemical-denatured hen egg-white lysozyme (HEWL) increased the antibacterial activity against Gram-negative bacteria, representing a novel candidate antimicrobial agent [15,16].

Urea is one of the most widely used chemical denaturing agents; it is used to understand the relationship between the structure and enzymatic function of proteins, as well as their conformational stability. It is known that urea increases the solubility of nonpolar molecules in water and prevents the formation of micelles. This effect is substantially more pronounced with regard to proteins [1]; urea acts both directly, by improving the solubility of hydrophobic residues, and indirectly, by disrupting the aqueous network structure that forms around the native protein [17,18]. Therefore, when the urea concentration increases, the protein increasingly loses its compact shape in favor of a flexible structure.

Porcine pepsin (molecular weight: 34.55 kDa) is a gastric aspartic proteinase that has been the subject of extensive study; pepsin is known to play an integral role in the digestive process of many vertebrates [19,20]. X-ray crystallography studies have shown that porcine pepsin consists of two structurally homologous domains, an N-terminal domain (residues 1–172), and a C-terminal domain (residues 173–326), and that its active binding site is located in the cleft between the domains; the secondary structures of these domains were found to consist almost entirely of β-sheets [19,20,21]. For pepsin to be activated, the catalytic site requires two aspartate residues, Asp32 and Asp215, one of which has to be protonated, and the other deprotonated [22]. The activation of these residues has been found to occur over a pH range of 1–5 [19,23]. Pepsin undergoes a conformational transition from a native (at acidic pH) to a denatured (at alkaline pH) state over a narrow pH range (between 6 and 7) [24,25,26,27]. The denaturation of porcine pepsin as a function of pH and temperature has been extensively studied in terms of its secondary structure and enzymatic activity using a variety of spectroscopic methods [24,25,26,27,28,29,30]. We previously studied the structure of porcine pepsin at various pH values in terms of their size and shape using small-angle X-ray scattering (SAXS) [31]. However, in order to obtain a detailed picture of its 3D structure and conformational stability, the analysis of the effect of the denaturing agent urea on the structure and enzymatic activity of pepsin is necessary.

While macromolecular X-ray crystallography and nuclear magnetic resonance (NMR) are yielding exponentially growing data on native protein structures, a relatively limited number of biophysical techniques can provide information on the denatured state that is defined as unstructured [2].

SAXS is a powerful method for the structural characterization of both ordered and disordered proteins in solution. It provides information regarding the sizes and shapes of proteins and complexes over a broad range of molecular sizes ranging from several kDa to GDa [32,33,34]. In particular, SAXS can comprehensively characterize dynamic processes in systems where macromolecular structures are evolving under varying experimental conditions (e.g., time, pH, pressure, or different reagents) [35,36,37,38]. In general, a highly purified monodisperse ideal sample without intermolecular interactions is essential for the reconstruction of 3D models from SAXS data. However, no additional labeling, crystallization, freezing, or chemical modification of the sample is needed, which makes SAXS universally applicable. SAXS is particularly suitable for studying flexible and disordered proteins, large biomolecular assemblies and oligomeric mixtures, and enzyme kinetics, which are difficult to investigate using other structural biology methods [35].

In this study, to obtain more information on the 3D structure of porcine pepsin and its conformational stability, and because of the importance of establishing the relationship between the structure and enzymatic activity of proteins, we investigated the effect of the denaturing agent urea on the structure of porcine pepsin using Raman spectroscopy and SAXS.

## 2. Materials and Methods

### 2.1. Materials and Sample Preparation

High-grade crystallized and lyophilized pepsin (3,690 units/mg), extracted from porcine stomach mucosa, was purchased from Sigma Aldrich (St. Louis, MO, USA), and its purity was confirmed using sodium dodecyl sulfate-polyacrylamide gel electrophoresis (SDS-PAGE) (Figure 1). Urea was purchased from Sigma Aldrich. Pepsin solutions were treated with a wide urea concentration range of 0–10 M. The samples were incubated at 25 °C in 20 mM sodium acetate solution (pH 4.5) and at the indicated concentrations of urea for 24 h according to a previous report [39], and then centrifuged at 12,000 rpm for 10 min at 4 °C and filtered through 0.20-μm pore size filter membranes before analysis. The protein concentration of the samples was determined based on the absorbance at 280 nm using a theoretical extinction coefficient calculated based on the amino acid sequence of the protein. However, the extinction coefficient of a protein is urea-concentration dependent. Please note that we did not use the concentration values of pepsin solutions with urea for data analyses, which were used exclusively to prepare a sample solution of a suitable concentration for SAXS measurement.

### 2.2. SDS-PAGE Analysis

Pepsin was dissolved in 20 mM sodium acetate buffer (pH 4.5), mixed with SDS at a ratio of 1:5, and incubated at 95 °C for 5 min. ACCU pre-stained protein marker was used as molecular weight size markers (T&I, Chuncheon-si, Gangwon-do, South Korea). The pepsin and molecular weight marker solutions were then subjected to 13% acrylamide separation gel and Mini PROTEAN Tetra system (Bio-Rad, Hercules, CA, USA). Coomassie brilliant blue R-250 staining reagent was purchased from Sigma Aldrich (St. Louis, MO, USA).

### 2.3. Asymmetrical Flow Field-Flow Fractionation with Multi-Angle Light Scattering (AF4-MALS)

The molecular weight of pepsin was determined using the Eclipse AF4-MALS detector system (Wyatt Technology, Santa Barbara, CA, USA) equipped with a standard channel (25 cm), 350-μm spacer, and regenerated cellulose membrane (10-kDa cutoff), and the DAWN Heleos II 18-angle MALS system (Wyatt Technology) was used for detection. A total volume of 50 μL protein solution at a concentration of 5.0 mg/mL was injected at a detector flow rate of 0.6 mL/min and a cross-flow rate of 2.5 mL/min. Data were analyzed using the Zimm model for fitting experimental light scattering data and graphed using EASI graph with a RI peak in ASTRA 6.1 software (Wyatt Technology).

### 2.4. Raman Spectroscopy

The vibrational spectra of the pepsin protein solutions at various urea concentrations in aqueous media were obtained using the RAMANforce dispersive Raman spectrometer (Nanophoton, Osaka, Japan) equipped with a 532-nm green laser source. The groove densities of the diffraction gratings were 300 and 1800 lines per mm. The Raman scatterings were recorded using an air-cooled front-illuminated charge-coupled device detector operating at −70 °C. The spectra were calibrated to 520 cm^−1^ as the representative peak position of a silicon standard sample, a capillary cell was used to fill the pepsin protein solutions, and data were collected by taking the average of 60 successive 1-s frames.

### 2.5. SAXS Measurements in Solution

SAXS measurements were carried out using the 4C SAXS II beamline (BL) [40] of the Pohang Light Source II (PLS II) with 3 GeV power at the Pohang University of Science and Technology (POSTECH), Pohang, Korea. A light source from an In-vacuum Undulator 20 (IVU20: 1.4 m length, 20 mm period) of the Pohang Light Source II storage ring was focused with a vertical focusing toroidal mirror coated with rhodium and monochromatized with a Si (111) double-crystal monochromator, yielding an X-ray beam wavelength of 0.734 Å. The X-ray beam size at the sample stage was 0.15 (V) × 0.24 (H) mm^2^. A 2D SX 165 charge-coupled detector (Rayonix, Evanston, IL, USA) was employed. A sample-to-detector distance of 4.00 m and 1.00 m were used for SAXS. The magnitude of scattering vector, *q* = (4π/*λ*) sinθ, was 0.1 nm^−1^ < *q* < 3.00 nm^−1^, where 2θ is the scattering angle, and *λ* is the wavelength of the X-ray beam source. The scattering angle was calibrated with polystyrene–*b*–polyethylene–*b*–polybutadiene–*b*–polystyrene (SEBS) block copolymer standard. A quartz capillary with an outside diameter of 1.5 mm and wall thickness of 0.01 mm was used as a solution sample cell. To obtain high-quality scattering data without any interparticle interference, solute concentrations of approximately 4.1–5.1 mg/mL were measured at a constant temperature of 25 °C in an FP50-HL refrigerated circulator (Julabo, Seelbach, Germany), as described in a previous report [31]. The SAXS data were collected in ten successive 10-s frames at a flow rate of 0.3 μL/s through a Microlab 600 advanced syringe pump (Hamilton, Reno, NV, USA). Data were normalized to the intensity of the transmitted beam and radially averaged. The scattering of specific buffer solutions was used as the experimental background. The *R*_g,G_ (radius of gyration) values were estimated from the scattering data using Guinier analysis [41]. The pair distance distribution p(r) function was obtained through the indirect Fourier transform method using the program GNOM [42]. The molecular mass (MM) was calculated from the protein volume and density [43].

### 2.6. Construction of 3D Structural Models

To reconstruct molecular models, the ab initio shape determination program GASBOR [44] was used. The theoretical SAXS curve of the atomic crystal structure was calculated using the CRYSOL program [45]. For the comparison of overall shapes and dimensions, ribbon diagrams of the atomic crystal models were superimposed onto reconstructed dummy residues models using the SUPCOMB program [46]. To select a subset of conformations from the theoretical random pool, the ensemble optimization method (EOM) program was utilized [47].

## 3. Results and Discussion

Prior to MALS and SAXS measurements, the purity of the pepsin samples was confirmed using 13% acrylamide separation gel, and bands were visualized using Coomassie brilliant blue R-250 staining reagent. Figure 1a shows the SDS-PAGE electropherogram of the pepsin molecules in the buffer solution in the absence of urea. As shown in Figure 1a, one strong band was observed at the 35–40 kDa marker, which is close to the theoretical molecular weight of pepsin (34.5 kDa). To corroborate this result and investigate the monodispersity of pepsin in the buffer solution in more detail, AF4-MALS measurement was performed. Figure 1b shows the AF4-MALS profile of pepsin in the buffer solution in the absence of urea; we observed a single distinct peak with no other oligomers or aggregates. The molecular mass of pepsin protein was estimated to be approximately 38.3 kDa, similar to its theoretical molecular mass (34.5 kDa). This result indicates that the pepsin molecules were present in a monodisperse, monomeric conformational state in terms of both overall size and molecular mass in solution.

To understand the structural changes in pepsin at the secondary structure level at various concentrations of the denaturing agent urea in aqueous media, Raman measurements were performed using a dispersive Raman spectrometer. The Raman spectra of the film and solution states of pepsin were distinct in terms of both peak position and broadness (data not shown). These Raman spectra showed not only intra-molecular information such as peptide secondary structure, but also peptide-environment interactions such as those between the peptide and the buffer solution and between the peptide and the denaturing agent. We observed representative Raman bands that indicated conformational changes in pepsin in aqueous media known as the amide I (1600–1800 cm^−1^) and amide III (1230–1400 cm^−1^) bands [48,49]. The amide I band (C=O stretch vibration) is most often characterized to clarify the secondary structures of proteins in solution, such as α-helix, β-sheet, and random-coiled conformations. However, in this region, detailed structural analysis was limited because of the strong Raman signals caused by the C=O vibrations of sodium acetate and urea in the buffer solution used in this study (data not shown). The spectral range of the amide III bands (H–N–C bending vibration) is suitable for observations without strong interference by unexpected Raman scattering. The amide III bands are also considered to be markers of secondary structures such as helices (260–1400 cm^−1^), β-sheets (1230–1240 cm^−1^), and random-coils (1240–1260 cm^−1^) [48,49,50,51]. Figure 2a shows the Raman spectra of pepsin in solution with the amide III bands as a function of urea concentration. A broad Raman band of approximately 1250 cm^−1^ could be classified into two types: β-sheet and random-coil conformations at 1231 cm^−1^ and 1246 cm^−1^, respectively. The intensity of the β-sheet band appeared to decrease gradually at 6–10 M urea when compared to those at 0–5 M urea. Conversely, there were no significant differences in the intensity of the random-coil band over the urea concentration range of 0–10 M. The amide III band, which is related to helical conformations, was prominent at 1341 and 1358 cm^−1^ and showed much more discrete double-peak patterns than those measured at 1231 and 1246 cm^−1^. It was also found that the Raman scattering patterns of pepsin molecules at 0–5 M urea were different from those measured at 6–10 M urea, similar to that observed for the amide III band at 1231 cm^−1^. For quantitative analysis, we attempted to calculate the integrated area of the amide III band at 1320–1370 cm^−1^. Figure 2b exhibits the integrated area of the band corresponding to the helical conformation as a function of urea concentration, and clearly shows that the value of integrated area of about 600 for pepsin at 0–5 M urea abruptly decreased to 350–400 at 6–7 M urea, and to approximately 300 at 8–10 M urea. Considering the Raman spectral data above, it can be predicted that pepsin molecules exist in a fully folded state at 0–5 M urea, are partially unfolded at 6–7 M urea and exist in a highly unfolded state at 8–10 M urea in terms of secondary structure. These results are similar to those of the unfolding process of pepsin as a function of the pH reported previously [30,39].

In order to investigate the effect of urea on the structural stability of pepsin, and to confirm the results of Raman analysis above, synchrotron SAXS measurements were performed at room temperature with 0–10 M urea. Our SAXS experimental results show that the pepsin structures formed during the initial incubation period are stable for 2 days, until the end of the SAXS experiment. In terms of structural stability, the 3D structure of pepsin before and after chemical denaturation as a function of urea concentration was studied quantitatively. The effect of urea concentration on the experimental X-ray scattering profiles of pepsin at pH 4.5 is shown in Figure 3. Overall, it appeared that the scattering profiles of the GASBOR ab initio envelope reasonably fit the experimental data for pepsin exposed to 0–10 M urea. We found that the experimental data for pepsin in the absence of urea were consistent with those calculated based on monomeric crystal structure of pepsin, indicating that pepsin without urea showed a very similar conformation to that observed in its crystal structure and that the absence of interparticle interference in our experimental conditions. Besides, there were also some similarities in the scattering patterns at 0–6 M urea. We recognized that as the urea concentration increased above 6 M to 10 M, the scattering patterns increasingly deviated from those measured at 0–6 M urea. This indicates that pepsin at 7–10 M urea underwent significant conformational changes, shifting from a globular form to a fully unfolded conformation due to urea-induced denaturation.

As the classical and most widely used structural parameter obtained from the SAXS data, the radius of gyration (*R*_g_) provides a measure of the overall size of biological macromolecules in solution [52]. The *R*_g_ can be defined as the average electron-density-weighted, root-mean-squared distance of the scatters from the center of density in the molecule [52]. The *R*_g_ value is smaller for proteins with a compact, globular state than for those with a partially and/or fully unfolded state, even though they have an equal number of amino acid residues [52]. The *R*_g_ can be calculated using the Guinier approximation using a small-angle region for *qR_g_* < 1.3 (hereafter, the radius of gyration obtained from the Guinier approximation is referred to as *R*_g,G_). Guinier plots of ln[*I*(*q*)] versus *q*^2^ of the scattering profiles of pepsin in solution at 0–10 M urea are displayed in Figure 4a,b. The scattering profiles plotted as ln[*I*(*q*)] vs. *q*^2^ showed linearity in the small *q*^2^ region. The *R*_g,G_ values of pepsin at different urea concentrations were determined based on the slope of the linear fit. The radius of gyration as a function of urea concentration was plotted, as shown in Figure 4c. In addition, we attempted to calculate pepsin molecular mass *MM*_SAXS_ from the SAXS intensity extrapolated to *q* = 0, *I*(0), the protein volume V (in Å) obtained from the invariant *Q*, and the average protein density of ρ_m_ = 0.83 × 10^−3^ kDa Å^−3^ [43]. Our data show that pepsin molecules exist in monomeric form in solution over a wide range of denaturant urea concentrations (0–10 M), despite some variations in MM values. The determined *R*_g,G_ and *MM* values are also listed in Table 1. The radius of gyration of pepsin in solution in the absence of urea was estimated to be 22.08 ± 0.22 Å, which almost exactly corresponded to that (21.34 ± 0.01 Å) of the crystal structure of monomeric pepsin (PDB:3PEP). The *R*_g,G_ values of pepsin at 0–6 M urea showed no significant differences, despite some variations in size. At concentrations of above 6 M urea, the *R*_g,G_ value increased significantly. Based on the resulting *R*_g,G_ values (Figure 4c and Table 1), we found that the onset urea concentration at which *R*_g,G_ begins to change significantly is approximately 7 M. The *R*_g,G_ value of pepsin at 7 M urea was 23.07 ± 0.79 Å, slightly larger than those measured at 0–6 M urea. The *R*_g,G_ values for pepsin at 8 and 9 M urea were estimated to be 32.66 ± 1.61 Å and 41.54 ± 3.43 Å, respectively, while that at 10 M urea was determined to be 50.01 ± 5.08 Å, the largest of all the measured *R*_g,G_ values. This result was considered to be due to complete unfolding as a consequence of the collapse of the secondary and tertiary structures. Fully unfolded proteins, because of their highly extended conformations, are characterized by a larger measured average size than globular proteins with tightly well-packed core domains [47,52]. Our comparison of the experimentally determined *R*_g,G_ value with that calculated from the theoretical models was used to characterize the unstructured nature of the protein. The predicted *R*_g_ value for the unstructured conformation of pepsin was directly calculated based on the number of amino acid residues using Flory’s equation [53]:*R*_g_ = *R*_0_*N^ν^*(1)
where *N* is the number of amino acid residues, *R*_0_ is a constant that depends on the persistence length of the peptide chain, and *ν* is an exponential scaling factor; for chemically denatured proteins we used the values *R*_0_ = 2.54 ± 0.01 and *ν* = 0.522 ± 0.01. The theoretically calculated *R*_g_ value, based on the fully unfolded state of pepsin with 326 amino acid residues, was estimated to be approximately 52.09 Å, which is consistent with that measured at 10 M urea (50.01 ± 5.08 Å). Through this observation, it was found that pepsin molecules exposed to 10 M urea have a fully extended, randomly coiled conformation in solution, owing to the complete destruction of the tightly well-packed core (sub)domain.

Generally, Kratky plots (*q*^2^*I*(*q*) as a function of *q*) are employed to qualitatively identify the denatured states of proteins in various denatured conditions and to distinguish them from their globular conformations; the Kratky plot can discriminate specific features of the scattering profiles of pepsin between its globular and fully unfolded conformations at different urea concentrations [52]. Typically, the scattering intensity from a solid body decays at high *q* values, with values of *I*(*q*) ~ 1/*q*^4^ with a bell-shaped pattern in the Kratky plot. Conversely, an ideal Gaussian chain has a 1/*q*^2^ asymptotic of *I*(*q*) and displays a plateau at larger *q* values. For an extended thin chain, the Kratky plot also presents a plateau over a specific range of *q*, which is followed by a monotonic increase [52]. Figure 5 shows the Kratky plots of SAXS data of pepsin at 0–10 M urea. At 0–5 M urea, we observed a distinct bell-shaped curve with a peak maximum at *q* = 0.08 in the small *q* region (*q* < 0.15). This distinct symmetrical peak pattern indicates that pepsin at 0–5 M urea had a compact, globular conformation, as expected. At 6 M urea, the Kratky plot also showed a distinct peak pattern at around *q* = 0.08 but was slightly asymmetrical compared to those at 0–5 M urea. This observation implies that pepsin underwent a weak urea-induced denaturation process at 6 M, though the overall size remained unchanged. At 7 M urea, the Kratky plot exhibited a weak peak maximum at around *q* = 0.1, with a shift toward high *q* region; the symmetrical or asymmetrical peak pattern observed at 0–6 M almost entirely disappeared. This result may have been because of the highly extended conformation due to the denaturation of the compact, globular pepsin structure. At 8–10 M urea, the Kratky plots showed an interesting scattering feature that differed markedly from those obtained at other urea concentrations. The peak maximum in the small *q* region was not present in the Kratky plot; the plot rose rapidly in the small *q* region and then increased gradually in the intermediate and high *q* regions. This pattern is typical of the Kratky-Porod chain [54], indicating that the secondary and tertiary structures of pepsin were almost completely destroyed at high urea concentrations, leading to a randomly coiled conformation.

In real space, the particle sample can be conveniently described using the distance distribution function *p*(*r*), which is a histogram of the distances between all possible pairs of atoms within a particle [52]. The *p*(*r*) function can be obtained from experimental scattering data using indirect Fourier transformation [55,56]. In some cases, it is more intuitive to interpret the structural properties rather than the scattering data itself. In particular, for compact globular particles, the *p*(*r*) shows a symmetrical bell-shaped pattern, whereas, for unfolded particles, it has an extended tail [52]. The radius of gyration can be calculated from the *p*(*r*) function (hereafter, referred to as *R*_g,p(r)_), which often provides a more reliable estimate than the Guinier approximation, especially in unstructured systems [57]. Figure 6a,b show the *p*(*r*) function for pepsin incubated with various urea concentrations based on an analysis of the experimental scattering data using indirect Fourier transformation. The areas under the curves were normalized to 1 for ease of comparison. The function provided the radius of gyration (*R*_g,p(r)_), which is based on the full scattering curve, as well as the maximum dimension (*D*_max_) of pepsin as the distance where the *p*(*r*) function approaches zero. The *R*_g,p(r)_ and *D*_max_ values obtained are summarized in Table 1. In addition, for an improved understanding of these size changes, the *p*(*r*) function-based radius of gyration and maximum dimension values as a function of urea concentration is plotted in Figure 6c. Overall, the resulting *R*_g,p(r)_ values were highly consistent with those from the Guinier approximation, despite several size variations. In particular, from the resulting *R*_g,p(r)_ values, we observed that the onset urea concentration leading to an increase in overall size is approximately 7 M, which is consistent with the estimates obtained from the Guinier approximation. We also found a relatively large difference between the *R*_g,p(r)_ and *R*_g,G_ values at both 9 and 10 M urea under fully denaturing conditions. Considering that the *R*_g,G_ is estimated from the restricted *q* range in the Guinier region, we expected that the *R*_g,p(r)_ based on the full scattering profile would be more reliable for pepsin molecules in a fully unfolded state under strongly denaturing conditions. We computed a theoretical *p*(*r*) function for the atomic coordinates of the crystal structure of the monomeric pepsin protein (PDB code 3PEP) and compared it with those measured for pepsin at 0–10 M urea. The *p*(*r*) functions for pepsin at 0–6 M urea exhibited a symmetrical peak pattern, which is characteristic of a compact globular conformation, and was considerably similar to that of the crystal structure (*R*_g,p(r)_ = 21.20 ± 0.19 Å, *D*_max_ = 67.0 Å) of pepsin. However, the *p*(*r*) functions for pepsin at 7–10 M urea clearly revealed an asymmetric form, with the maximum dimensions shifted toward the long-distance region compared to those measured at 0–6 M. The *D*_max_ value increased in the following order: 7 M urea (117.0 Å) < 8 M urea (129.0 Å) < 9 M urea (197.0 Å) ≤ 10 M urea (198.0 Å), indicating that the conformation of pepsin deviated considerably at increased urea concentrations higher than 6 M.

The correlations between the Kratky plot analysis, the *p*(*r*) function analysis, and the molecular structure were further examined by reconstructing ab initio structural models from the X-ray scattering data using GASBOR. To improve the reliability of the final solution models, multiple runs of GASBOR were used, and the most probable models among them were selected. Figure 7 shows the structural models reconstructed for pepsin in solution at 0–10 M urea. As described in the *p*(*r*) function analysis, the structural model for pepsin at 0 M urea appeared to be close to that of the monomeric crystal structure in terms of overall shape and dimension. The structural models for pepsin at 1–5 M urea showed a compact globular conformation, similar in overall shape to that at 0 M urea. At 6 M urea, the structural model showed a bent conformation along the horizontal axis in the central region between N- and C-terminal domains, slightly different to those at 0–5 M urea, but showing minimal differences in size.

It is interesting to note that based on the Kratky plot in Figure 5, pepsin should be present in a partially denatured and compact globular conformational state at 6 M urea. According to this solution model, we identified an intermediate pepsin conformation that was dominant during an early stage of denaturation. Furthermore, such a subtle conformational change may provide clues regarding the reduced enzymatic activity of pepsin under denaturing conditions, taking into account the activity result reported previously [58]. The structural model for pepsin at 7 M urea revealed a highly extended and linear conformation along the vertical axis. These results were consistent with the Kratky plot and *p*(*r*) function analyses results, which showed that pepsin at 7 M urea is present in a highly unfolded and extended conformational state. Similarly, the structural model at 8 M urea was more extended than that at 7 M urea. The structural models for pepsin at 9 and 10 M urea adopted a fully unfolded and randomly coiled conformation that was significantly extended along the vertical axis, as confirmed by the results of the Kratky plot and *p*(*r*) function analyses. However, it was suspected that pepsin molecules at high urea concentrations might coexist in multiple conformations with conformational polydispersity under highly denatured conditions.

Therefore, to confirm that the GASBOR-derived reconstructed model for pepsin at 10 M urea was a representative and reliable solution model for any interpretations presented, we attempted to generate an ensemble of possible conformations of pepsin and select a subset of conformations that best fit the experimental data for pepsin at 10 M urea, with a random sequence designation, using the EOM program (Figure 8). Interestingly, it was found that the EOM-derived subset of conformations showed randomly coiled conformations similar in overall shape to the GASBOR-derived dummy residue model for pepsin under the same conditions (Figure 9). Therefore, the GASBOR ab initio structural models for pepsin at high urea concentrations were confirmed to be trustworthy solution models that appropriately represented the fully unfolded conformations of pepsin under highly denatured conditions. Besides, we observed that pepsin was present in soluble submicron aggregates without precipitation when the pepsin solution, including 10 M urea, was diluted 10-fold (1 M) with a buffer solution in the absence of urea (data not shown). Collectively, from these results, we observed that pepsin is structurally dynamic over a wide range of urea concentrations, adopting several forms ranging from the crystal structure-like folded state to randomly coiled conformations such as the Kratky-Porod chain.

## 4. Conclusions

To our knowledge, this study has successfully demonstrated, for the first time, the 3D structure and structural transitions of porcine pepsin in solution at 0–10 M urea using SAXS analysis. This information is necessary for understanding the correlation between enzymatic activity and denaturation in pepsin, and for interpreting the conformational changes in the secondary structure measured by various spectroscopy techniques. In order to confirm the purity and monodispersity of the pepsin solution, SDS-PAGE and MALS measurements were performed. Based on Raman spectroscopy, it was determined that the urea concentration at which changes in the secondary structure of pepsin began to occur was approximately 6 M. Based on complementary analyses of the SAXS data, such as Guinier approximation, Kratky plot, *p*(*r*) function, and reconstructed model analyses, we demonstrated that pepsin undergoes conformational changes from its crystal structure-like compact globular state (0–5 M urea) to an initially denatured, intermediate form (6 M urea), and finally to a highly extended, randomly coiled conformation (7–10 M urea). The 3D structural models that adequately represented the conformations of pepsin under varying denaturing conditions were reconstructed using the ab initio shape determination method. Overall, the results of SAXS analysis were consistent with the Raman results, showing that the changes in the secondary structure of pepsin resulted in changes in its overall shape and dimensions. We believe that our reconstructed 3D solution models offer useful insights into the relationship between the enzymatic activity and conformation of pepsin under various urea concentrations. Besides, we expect that the structural evidence and analytical methods presented here will contribute to the establishment of correlations between the functional activity and structure of similar enzymatic proteins under various denaturing conditions.

## Figures and Tables

**Figure 1 polymers-11-02104-f001:**
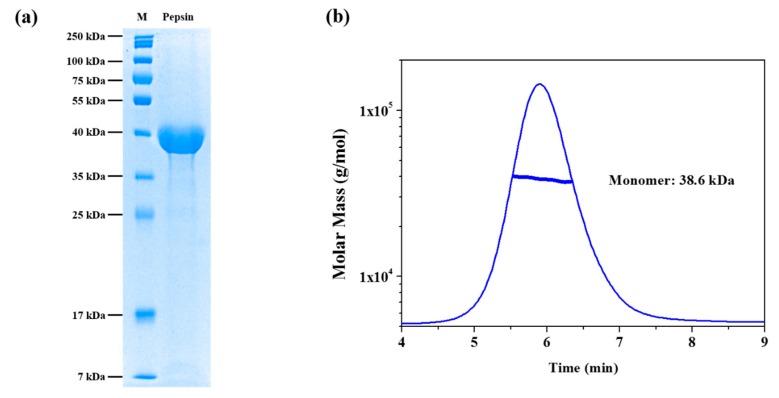
Purification and asymmetrical flow field flow fractionation with multi-angle light scattering (AF4-MALS) of pepsin in solution. (**a**) SDS-PAGE electropherogram of pepsin in solution in the absence of denaturant. M: molecular weight size marker. (**b**) Asymmetrical flow field-flow fractionation coupled with multi-angle light scattering (AF4-MALS) for pepsin in solution in the absence of denaturant. The thick line represents the determined molecular weight according to the Zimm model.

**Figure 2 polymers-11-02104-f002:**
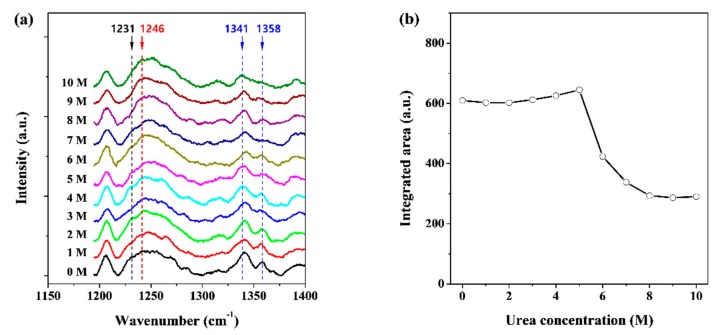
(**a**) Raman spectra of pepsin solutions as a function of urea concentration, which was measured at room temperature (approximately 25 °C) and (**b**) effect of increasing urea concentrations on the integrated area of amide III bands in the helical conformation.

**Figure 3 polymers-11-02104-f003:**
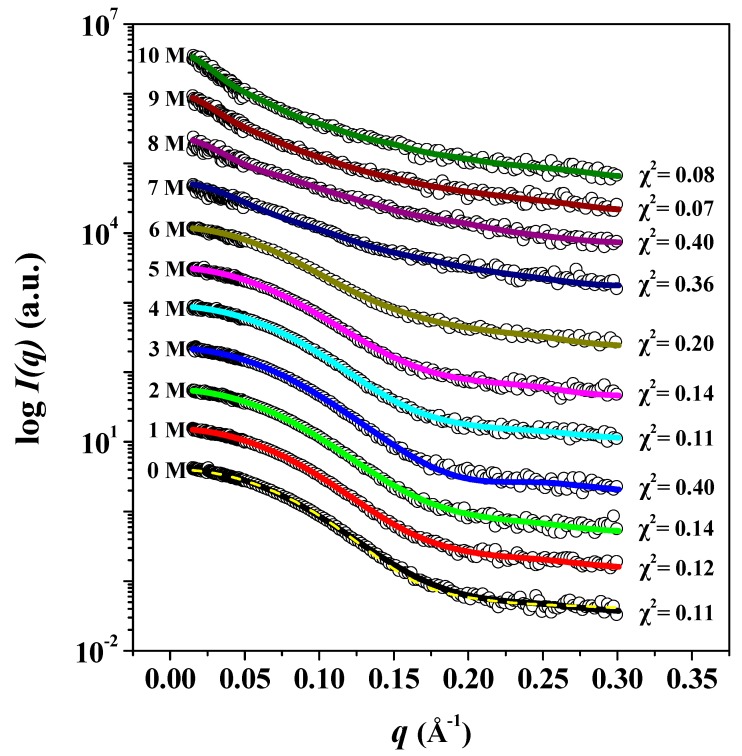
*I*(*q*) versus *q* as linear-log plots of pepsin in solution as a function of urea concentration. Open shapes represent the experimental data, and solid lines represent the X-ray scattering profiles obtained from the dummy residue models using GASBOR. The dashed yellow line shows the theoretical SAXS curve calculated from the crystal structure of monomeric pepsin (PDB:3PEP) (χ^2^ = 0.170). For clarity, each plot is shifted along the log *I*(*q*) axis.

**Figure 4 polymers-11-02104-f004:**
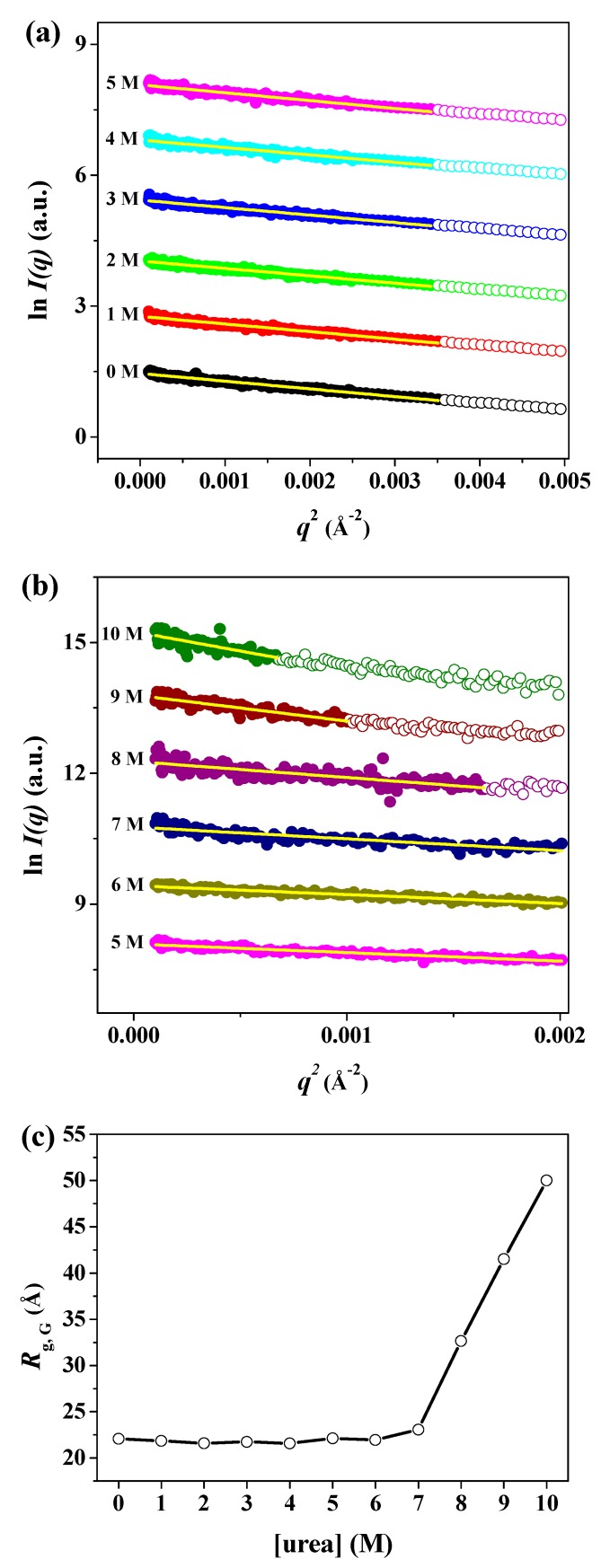
Guinier plots of X-ray scattering profiles of pepsin in solution at various urea concentrations. (**a**,**b**) Guinier plots of ln[*I*(*q*)] versus *q*^2^ of the scattering profiles of pepsin protein molecules in solutions at (**a**) 0–5 M and (**b**) 5–10 M urea. The Guinier fits (yellow lines) were obtained from the linearity of the scattering data in the *q*^2^ region for *qR_g_* < 1.3. Open shapes represent data beyond the Guinier region. For clarity, each plot is shifted along the ln *I*(*q*) axis. (**c**) Effect of urea on the experimental radius of gyration of pepsin protein at pH 4.5. The solid line is a guide for ease of comparison.

**Figure 5 polymers-11-02104-f005:**
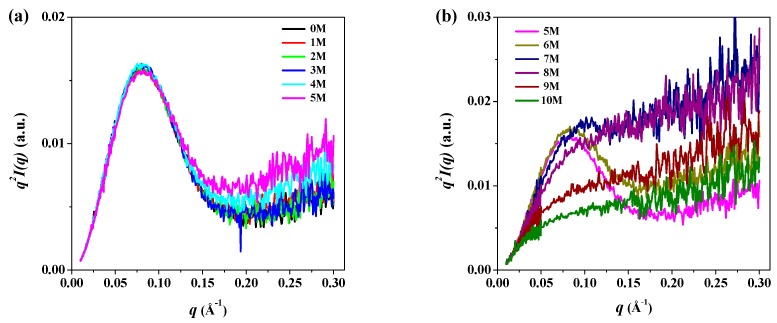
Kratky plots of *q*^2^*I*(*q*) versus *q* of the scattering profiles of pepsin in solution with (**a**) 0–5 M and (**b**) 5–10 M urea.

**Figure 6 polymers-11-02104-f006:**
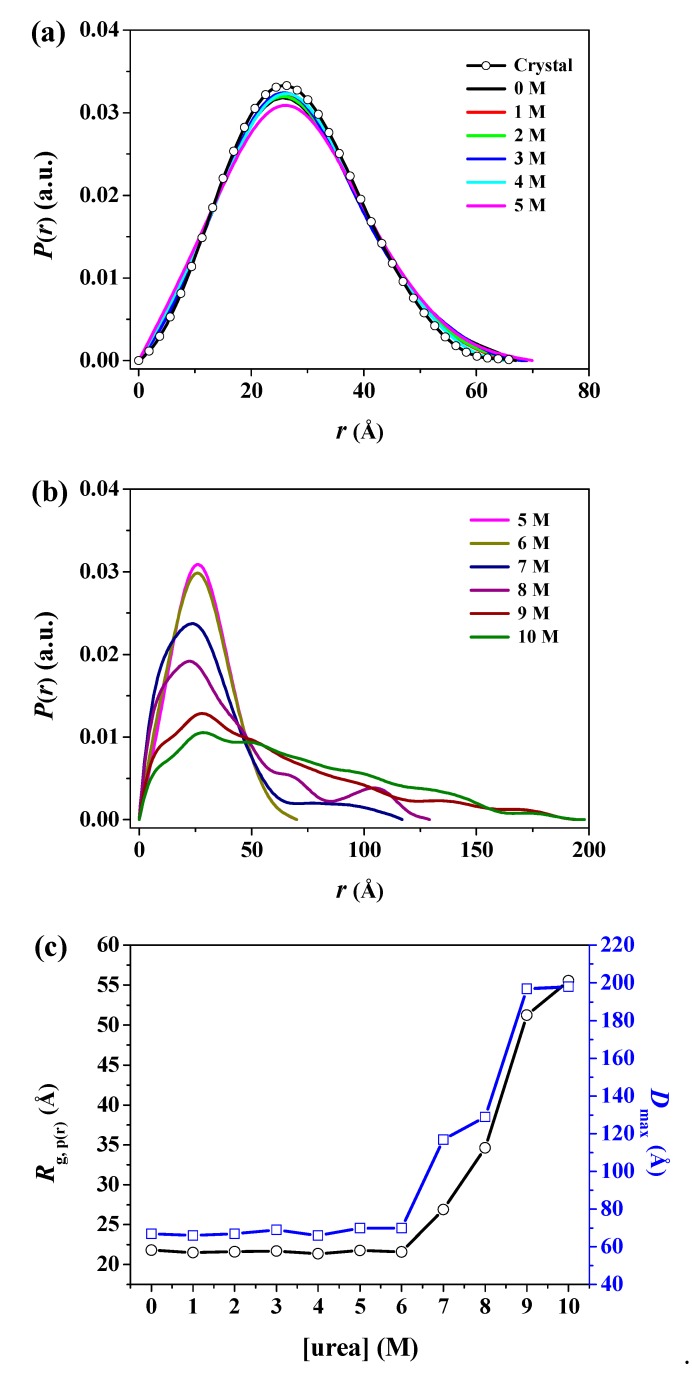
The pair distance distribution *p*(*r*) functions of pepsin in solution with various urea concentrations. (**a**,**b**) *P*(*r*) versus *r* profiles of pepsin in solution with (**a**) 0–5 M and (**b**) 5–10 M urea as a function of urea concentration, based on an analysis of the experimental SAXS data through GNOM. The areas under the curves were normalized to equal areas for ease of comparison. (**c**) Effect of urea on the *p*(*r*)-based radius of gyration (*R*_g,p(r)_) and maximum dimension (*D*_max_) of pepsin at pH 4.5. The solid line is a guide for ease of comparison.

**Figure 7 polymers-11-02104-f007:**
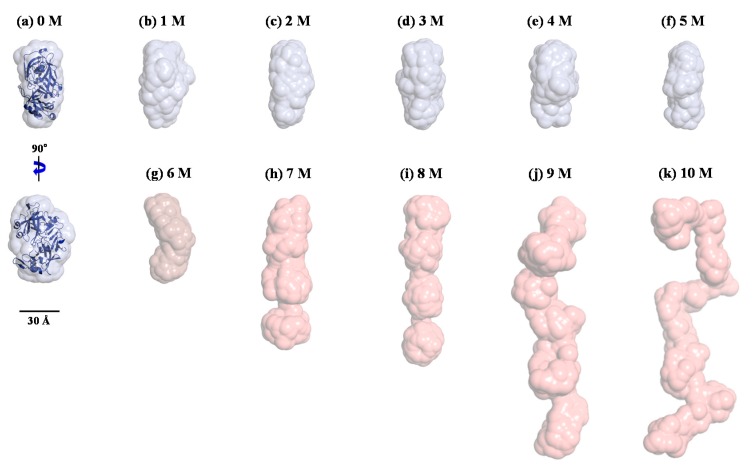
Structural models of pepsin in solution under various denaturing conditions with 0–10 M urea (**a–k**, respectively) using the ab initio shape method program GASBOR. Surface rendering in the structural model was achieved using the program PyMOL. For the comparison of overall shapes and dimensions, the ribbon diagrams of the atomic crystal structure of pepsin were superimposed on the reconstructed dummy residues models using SUPCOMB (NSD = 1.802).

**Figure 8 polymers-11-02104-f008:**
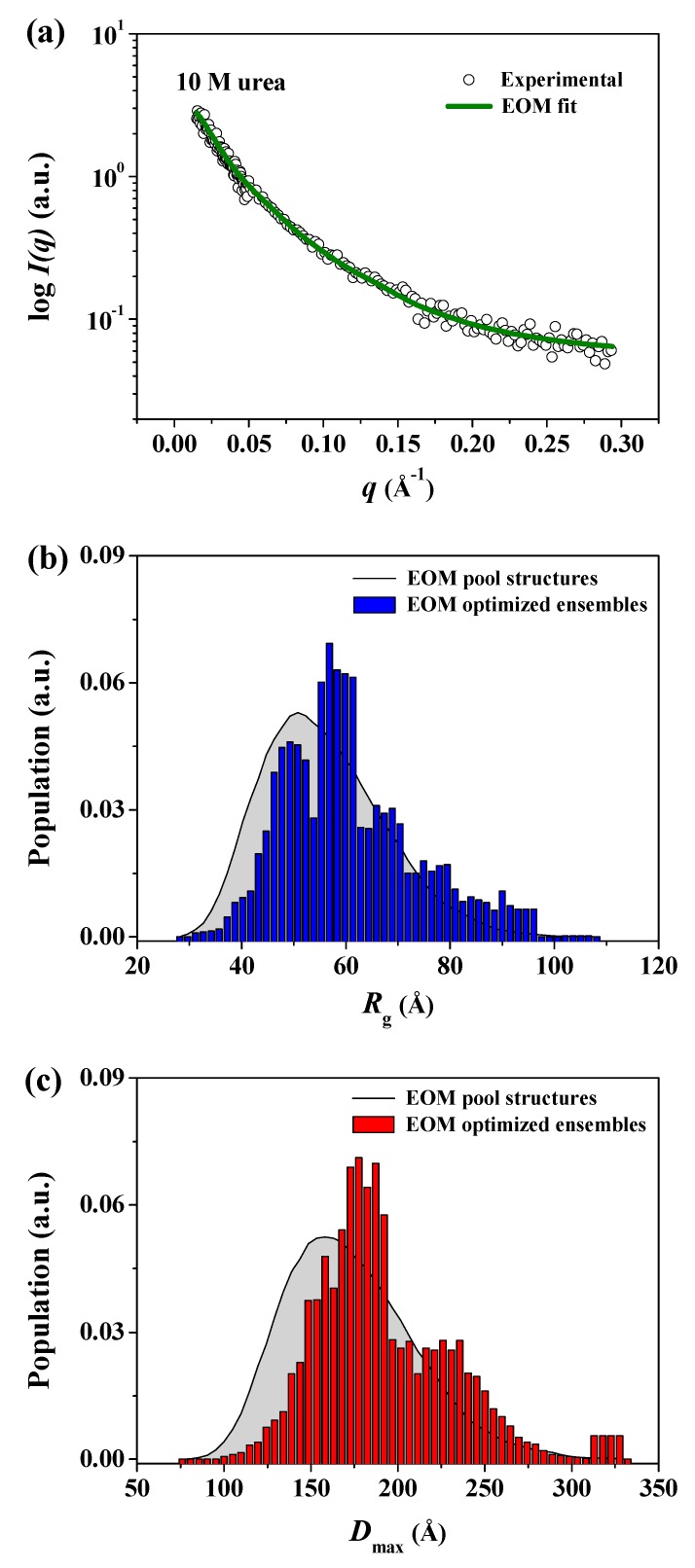
Distributions of *R*_g_ and *D*_max_ pools and selected ensembles of pepsin using ensemble optimization method (EOM). (**a**) X-ray scattering profile of pepsin in solution with 10 M urea. Open shapes represent the experimental data, and the solid line shows the fit obtained from EOM. The discrepancy (χ^2^) between the experimental and theoretical curves was calculated as 0.079. (**b**,**c**) Size distribution functions for pepsin in solution at 10 M urea using the EOM program. (**b**) *R*_g_ parameter distribution, (**c**) *D*_max_ parameter distribution. The distributions corresponding to a large pool of 10,000 randomized conformations are shown as solid gray lines. The distributions of optimized ensembles of *N* = 50 samples selected by the generic algorithm are shown as blue (*R*_g_) and red (*D*_max_) bars, respectively. R_flex (random)_/R_sigma_ = ~86.53% (~85.90%)/1.10.

**Figure 9 polymers-11-02104-f009:**
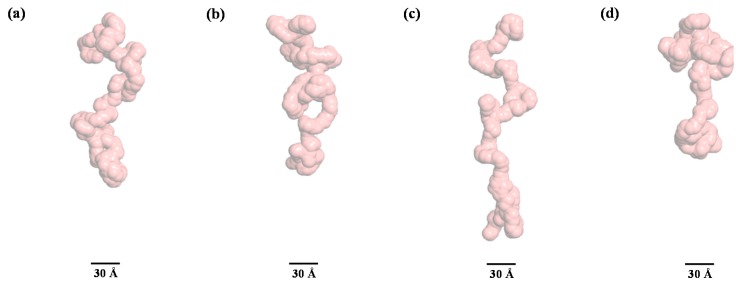
A subset of fully unfolded conformations selected from the generated ensembles for pepsin at 10 M urea through the ensemble optimization method. (**a**) *R*_g_ = 59.31 Å, *D*_max_ = 184.37 Å, volume fraction = 36%, (**b**) *R*_g_ = 49.68 Å, *D*_max_ = 168.99 Å, volume fraction = 29%, (**c**) *R*_g_ = 78.23 Å, *D*_max_ = 246.74 Å, volume fraction = 21%, (**d**) *R*_g_ = 48.76 Å, *D*_max_ = 148.60 Å, volume fraction = 14%.

**Table 1 polymers-11-02104-t001:** Structural parameters obtained of pepsin at different urea concentrations.

Sample	*R*_g,G_*^a^* (Å)	*R*_g,p(r)_*^b^* (Å)	*R*_g,p(r)_/*R*_g,0M_^*c*^	*D*_max_*^d^* (Å)	*MM*_SAXS_*^e^* (kDa)
Crystal	21.34 ± 0.01	21.20 ± 0.19	0.97	67.0	-
0 M	22.08 ± 0.22	21.79 ± 0.13	1.00	67.0	39.04
1 M	21.84 ± 0.23	21.50 ± 0.15	0.99	66.0	35.48
2 M	21.59 ± 0.24	21.60 ± 0.15	0.99	67.0	36.94
3 M	21.73 ± 0.26	21.68 ± 0.21	0.99	69.0	37.08
4 M	21.58 ± 0.27	21.34 ± 0.20	0.98	66.0	31.33
5 M	22.12 ± 0.31	21.76 ± 0.21	0.99	69.9	28.39
6 M	21.96 ± 0.38	21.57 ± 0.24	0.99	70.0	33.67
7 M	23.07 ± 0.79	26.89 ± 2.54	1.23	117.0	20.85
8 M	32.66 ± 1.61	34.62 ± 4.25	1.59	129.0	25.89
9 M	41.54 ± 3.43	51.25 ± 5.98	2.35	197.0	39.67
10 M	50.01 ± 5.08	55.52 ± 6.41	2.55	198.0	34.18

^a^*R*_g,G_ (radius of gyration) was obtained from the scattering data by Guinier analysis. ^b^
*R*_g,p(r)_ (radius of gyration) was obtained from the *p*(*r*) function using GNOM. ^c^
*R*_g,p(r)_/*R*_g,0M_ is the ratio of the *R*_g,p(r)_ value at different urea concentrations relative to that of the folded state at 0 M urea. ^d^
*D*_max_ (maximum dimension) was obtained from the *p*(*r*) function using GNOM. ^e^
*MM*_SAXS_ (molecular mass) was calculated from the protein volume and density.

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
