# Peer review of "Chemically Denatured Structures of Porcine Pepsin using Small-Angle X-ray Scattering"

_polymers, 2019, doi:10.3390/polym11122104_

Round 1

Reviewer 1 Report

Re: ms. "Chemically Denatured Structures of Porcine Pepsin using Small-Angle X-ray Scattering" by Rho et al.

The authors characterized the shapes, the sizes, the secondary structures of porcine pepsin at 0-10 M Urea using a small angle X-ray scattering (SAXS) and a Raman scattering. They found that the global structure of porcine pepsin changed at > 6M Urea. The conclusion was based on well-established analyses for SAXS such as Guinier analysis or a distance distribution function, P(r), analysis. The unfolded structure could be flexible. Thus, ensemble optimization (EOM) analysis helps to understand the conformations at 10 M Urea. These data should be recorded and are worthy of publication posterior to revision. As revision, I have questions and suggestions as follows.

<Major question (to be clarified)>
1. Throughout the manuscript, I cannot figure out that the protein solutions are aggregates-free except 0 M Urea solution (as depicted in Fig.1). The pepsin envelops and volumes at 7-10 M Urea in Fig.7 look larger than those at 0-6 M Urea. Were the pepsin molecules at 7-10 M mono-dispersed? To prove the monodispersity, the authors could present the molecular weights calculated by I(0), zero-angle X-ray scattering intensity, or light scattering intensity. Any other methods to prove monodipersity are acceptable such as size exclusion chromatography. Given power-law behavior (eq.1) of the pepsin chain implied that the pepsin at 10 M Urea was monomeric (yet remains to be unclear). In Fig.6, the authors normalized the areas of P(r) functions to have the same area. But, such normalization possibly covers up the effect of aggregation contaminating. The authors should reject the possible contamination of the aggregates.

2. p.2, Line92-94: An extinction coefficient of a protein is urea-concentration dependent. Did the authors deal with this? Or Could the authors explain more how the protein concentrations at 0-10 M Urea were determined?

3. p.3, Line129-130: The protein concentrations were 4.1-5.1 mg/mL. Did the authors check the concentration-dependence of the SAXS profile? 4.1-5.1 mg/mL possibly gives the structure factor, the interference, that modifies the SAXS profile, especially in the small angle region. Or could the author present some evidence of no interference effect on SAXS?

4. p.2, Line91: The pepsin was incubated with Urea for 24 hours. Does the structural change of the pepsin complete within 24 hours? Did the authors check the time dependence of urea-induced unfolding?

5. p.16, Line398-400: The Raman scattering indicated the disruption of the secondary structure at ~6 M Urea, while the SAXS profiles at ~6 M and at >~7M Urea represented the globular shape and the chain-like shapes, respectively. Does the pepsin have a globular shape with a lack of the secondary structure at ~6M Urea?

<Minor questions/suggestion>
1. p.2, Line63-64: "However, the urea‐induced chemical denaturation of porcine pepsin has received little attention in terms of its 3D structure" Why did this phenomenon receive little attention?

2. p.3, Line108: vibration --> vibrational

3. p.3, Line127: Should "b" in "-b-" be italic? (Could you check it?)

4. p.5, Figure 2: The wavenumbers of "1231", "1246", "1341", and "1358" indicated in the Fig could be helpful for readers. The temperature should be indicated in the figure caption (room temperature?).

5. p.5, Line193-194: The term "partially unfolded" in this sentence presumed that the pepsin unfolds via intermediate(s). Do the authors have some evidence of the non-two-state transition of the pepsin?

6. p.6, Figure 3: It may be better that each value of chi^2 is shown in the figure. "the crystal structure of monomeric pepsin" --> "the crystal structure of monomeric pepsin (PDB: 3PEP)"

7. p.6, Line 227: Two kinds of Rg (Rg,G and Rg,P(r)) were presented in this manuscript. It would be better to see the definition of Rg,G, here in the document.

8. p.10, Line286: Could the authors present the reference(s) of "Kratky-Porod chain"?

9. p.10, Figure5: Why did the scattering (8-10 M Urea) have discontinuous larger noises at q = ~0.05 angstroms-1?

10. p.10, Line307-310: Could you explain more about "supporting..."? It is not easy to understand "supporting..." based on the difference Rg,G and Rg,P(r).

11. p.13, Figure7: Did the authors consider the electron density difference between water and urea solution for the GASBOR modeling calculation?

12. p.14, Line369-372: This phenomenon looks interesting. Presenting the data would be fruitful.

Reviewer 2 Report

The present work uses different analytical techniques to characterize the denatured form of porcine pepsin . The work needs further revisions before being accepted by the Polymers magazine (MDPI).

A paragraph could be added that talks about the denatured form of proteins and their advantages to modify and / or enhance their biological activities, for example, lysozyme from egg white and bovine lactoferrin. The following articles could be cited as references:

Vilcacundo, R., Méndez, P., Reyes, W., Romero, H., Pinto, A., & Carrillo, W. (2018). Antibacterial Activity of Hen Egg White Lysozyme Denatured by Thermal and Chemical Treatments. Scientia Pharmaceutica86(4), 48.

Cegielska-Radziejewska, R., Lesnierowski, G., & Kijowski, J. (2009). Antibacterial activity of hen egg white lysozyme modified by thermochemical technique. European Food Research and Technology228(5), 841-845.

Please, define the term SDS-PAGE in Material and Methods

Indicate pepsin units

In materials and methods it is indicated that pepsin was denatured with 0-10 M urea at pH 4.5. Enzymes usually behave differently depending on the pH of the solution. Pepsin has an activity range between pH 2.0 -pH 5.0 approximately. Why weren't other denaturation pHs tested?

The SDS-PAGE analysis section should be extended, for example, the brand of the equipment used and the general conditions such as% acrylamides, coomassie staining, standard marker should be indicated. Indicate if mercaptoethanol was used in the assay. Some of these data are in the results section and should be recorded in materials and methods.

Pepsin is shown in Figure 1a of SDS-PAGE. Is pepsin denatured with urea 0, 1, 10 M. At what concentration?

Remove the molecular weights from the marker from the figure legend: (from top to bottom: 250, 170, 130, 100, 75, 55, 40, 35, 25, 17, and 7 kDa).

The graphical abstract must be improved. it could be indicated that the molecule that was worked with is pepsin and that it is its denatured form

Round 2

Reviewer 1 Report

[Major question]
Point 1: I believe that the protein solution at 0-10 M urea would be aggregation-free according to the explanation in response 1 by the authors. Potential readers of this paper would suspect that the protein aggregated, as I did. Thus, I recommend that the authors document rejection of the possible aggregation; the current revised manuscript does not describe that. As for I(0) and MW, I agree with the authors' comment of the uncertainty of the protein volume in urea solutions. On the other hand, the difference in average electron density between the protein and the urea solution is theoretically available. Thus, could the authors estimate MW assuming that the partial specific volume(PSV) of the protein is urea-independent? Indeed, the dependence of PSV of a protein on urea concentration looks small; e.g., Lapanje et al., Croatica Chemica Acta, 43, 65 (1971). Unfortunately, as the author indicated, the concentrations of the protein in urea solutions would not be exact. Nevertheless, showing that the estimated MW is monomeric MW rather than dimeric MW would be sufficient. Could the authors show that? The DLS result would be supporting. But, Volume% is not straightforward. Intensity% would be better because both the DLS and SAXS monitored the scattering: a tiny amount of the aggregates influences the SAXS as well as DLS of Intensity%. In addition, a urea solution possibly suppresses the aggregation of proteins as the authors' finding at 10 M urea. This fact is also supporting.

Point 2: As the authors said, I am sure that the given protein concentration (4.1-5.1 mg/mL) is a rough estimation because of the lack of the exact extinct coefficients in urea solutions. I recommend that the authors insert some notes on that in the manuscript.

Point 3 and 4: The authors' explanation is sufficient. Could the authors describe that in the manuscript?

[Minor question]
Point 1: The authors' explanation is fruitful. I intend to recommend, accordingly, the original sentence "However, the urea ... its 3D structure" is replaced by this explanation in the response.

Point 5: As the authors said, the unfolding process would be multi-step. Some references regarding the equilibrium unfolding process (not kinetic process) could be indicated (this is my suggestion).

Point 11: GASBOR has the parameter, "Contrast of the hydration layer relative to the solvent," the value of which would change upon the addition of urea. Is this negligible?

Author Response

Our manuscript has been revised according to the reviewer’s comments. Please refer to our point-by-point responses to the reviewer’s comments, listed in the following pages.

[Major question]

Point 1: I believe that the protein solution at 0-10 M urea would be aggregation-free according to the explanation in response 1 by the authors. Potential readers of this paper would suspect that the protein aggregated, as I did. Thus, I recommend that the authors document rejection of the possible aggregation; the current revised manuscript does not describe that. As for I(0) and MW, I agree with the authors' comment of the uncertainty of the protein volume in urea solutions. On the other hand, the difference in average electron density between the protein and the urea solution is theoretically available. Thus, could the authors estimate MW assuming that the partial specific volume(PSV) of the protein is urea-independent? Indeed, the dependence of PSV of a protein on urea concentration looks small; e.g., Lapanje et al., Croatica Chemica Acta, 43, 65 (1971). Unfortunately, as the author indicated, the concentrations of the protein in urea solutions would not be exact. Nevertheless, showing that the estimated MW is monomeric MW rather than dimeric MW would be sufficient. Could the authors show that? The DLS result would be supporting. But, Volume% is not straightforward. Intensity% would be better because both the DLS and SAXS monitored the scattering: a tiny amount of the aggregates influences the SAXS as well as DLS of Intensity%. In addition, a urea solution possibly suppresses the aggregation of proteins as the authors' finding at 10 M urea. This fact is also supporting.

Response 1: The reviewer’s valuable comments are genuinely appreciated. As the reviewer suggested, we calculated the MM values and added the determined values in Table 1 of the revised manuscript. The corresponding revisions are shown in blue font.

Point 2: As the authors said, I am sure that the given protein concentration (4.1-5.1 mg/mL) is a rough estimation because of the lack of the exact extinct coefficients in urea solutions. I recommend that the authors insert some notes on that in the manuscript.

Response 2: We thank the reviewer for their valuable comment. According to the suggestion, we added some notes in the Materials and Methods section of the revised manuscript. The corresponding revisions are shown in blue font.

Point 3 and 4: The authors' explanation is sufficient. Could the authors describe that in the manuscript?

Response 3: The reviewer’s valuable comment is very much appreciated. The corresponding revisions are shown in blue font.

[Minor questions]

Point 1: The authors' explanation is fruitful. I intend to recommend, accordingly, the original sentence "However, the urea ... its 3D structure" is replaced by this explanation in the response.

Response 1: We thank the reviewer for the comment. According to the suggestion, the required correction was made and shown in blue font.

Point 5: As the authors said, the unfolding process would be multi-step. Some references regarding the equilibrium unfolding process (not kinetic process) could be indicated (this is my suggestion).

Response 5: We thank the reviewer for the comment. According to the, additional references were added. The corresponding revisions are shown in blue font.

Point 11: GASBOR has the parameter, "Contrast of the hydration layer relative to the solvent," the value of which would change upon the addition of urea. Is this negligible?

Response 11: We thank the reviewer for the valuable comment. Here, the contrast of the hydration layer relative to the solvent refers to the difference of electron density of the water solvent layer bound to the protein surface relative to the water (bulk) solvent at relatively large distance from the protein. The program GASBOR assumes that the bound water molecules would be 10 % denser than the bulk water molecules (δρb = 30 e/nm3). The author predicts that this value for pepsin in solution at various urea concentrations will be nearly the same, and thus negligible. However, recent studies have shown that for charged biomolecules under ion cloud conditions, the contribution from the hydration layer and ion cloud can influence the experimental X-ray scattering pattern, complicating the interpretation of the data.

Reviewer 2 Report

The study with title: Chemically Denatured Structures of Porcine Pepsin
3 using Small-Angle X-ray Scattering in the present form is suitable for publication in the Polymers MDPI journal. Thanks to the authors for accepting the suggestions to improve the quality of the work.

Author Response

Thank you for the comments